# The role of a structured community health worker network in achieving malaria elimination goals in the Dominican Republic: An impact evaluation amid COVID-19 disruptions

**Nicole Michelén Ströfer**[1]*, **Natalia Tejada Bueno**[1], **María Yinet Santos Félix**[2], **Jose Luis Cruz Raposo**[2], **Justin T. Lana**[1], **Valérian Turbé**[1], **Harriet G. Napier**[1], **Justin M. Cohen**[1]

1 Clinton Health Access Initiative, Boston, Massachusetts, United States of America, 2 Ministerio de Salud Pública y Asistencia Social, Santo Domingo, Dominican Republic

* michelenstrofernicole@gmail.com

**Data Availability Statement:** All relevant data for this study are publicly available from the figshare

## Abstract

Community health workers (CHWs) can play a key role in malaria elimination through expanding access to malaria testing and treatment. The Dominican Republic, a low endemic country committed to eliminate malaria by 2025, implemented a structured malaria CHW network in the three main urban foci of Greater Santo Domingo. This research quantifies the networks' contributions towards malaria elimination from its implementation in 2019 until 2022. The study highlights the impact of COVID-19 disruptions on CHWs' performance and explores the network's ability to bounce back from health systems' shocks. The performance of the malaria CHW network was evaluated using weekly data collected from CHWs and routinely collected surveillance data from the Ministry of Public Health (MoH). We assess performance of CHWs by analyzing key variables including (1) reporting compliance, (2) household visitation rates, (3) malaria rapid diagnostic tests performed, (4) malaria cases detected, and (5) time between symptom onset and malaria diagnosis. To evaluate the impact of COVID-19 on the network, CHW's performance indicators are evaluated across three time periods (prior to, during, and after COVID-19 interruptions). Over the evaluation period, reported malaria cases in study foci decreased from 1,243 cases in 2019 to 6 cases in 2022. CHWs diagnosed and treated over 43% of malaria cases in study foci before COVID-19 interruptions and 14% during interruptions. 83% of cases detected by CHWs were detected through active case detection, with 17% detected passively. CHWs detected malaria cases and initiated treatment 1.5 days earlier than health facilities and MoH personnel performing active case detection. This evaluation provides evidence that a structured CHW network with clearly defined responsibilities and management protocol can help curb local malaria transmission. It adds to a growing body of research on the feasibility and benefits of CHW-led proactive household visitation.

repository (https://doi.org/10.6084/m9.figshare.26124820.v1).

**Funding:** This work was supported, in whole or in part, by the Bill & Melinda Gates Foundation INV-002736. Under the grant conditions of the Foundation, a Creative Commons Attribution 4.0 Generic License has already been assigned to the Author Accepted Manuscript version that might arise from this submission. The Clinton Health Access Initiative, with funds from the Bill and Melinda Gates Foundation, covered the cost of SurveyCTO, the platform used to collect CHW data, CHW initial trainings and implementation cost. The Regional Malaria Elimination Initiative covered the cost of CHW stipend since their implementation. The Bill & Melinda Gates Foundation and the Regional Malaria Elimination Initiative had no role in study design, data collection and analysis, decision to publish, or preparation of the manuscript.

**Competing interests:** The authors have declared that no competing interests exist.

## Introduction

CHWs play a key role in the global fight against malaria, primarily through expanding access to malaria testing and treatment in communities underserved or poorly reached by national health systems. In 2019, the Dominican Republic (DR), one of two malaria endemic Caribbean nations, signed the Regional Malaria Elimination Initiative (RMEI) pledge, demonstrating its commitment to eliminate malaria by 2020 (a date that was later postponed to 2025) [1]. The National Malaria Program (NMP) built its elimination programming around the Panamerican Health Organization/ World Health Organization (PAHO/WHO) recommended strategy: Diagnostics, Treatment, Investigation, and Response (DTI-R) [2]. This strategy emphasizes that malaria infected individuals should access diagnosis and treatment within the shortest possible time [2].

In line with the DTI-R strategy and the country's commitment to eliminating malaria, the NMP set up a CHW network to increase access to timely treatment and diagnosis in high-risk communities. Community integration for malaria elimination had been considered for decades, predating the establishment of CHWs networks in the country; however, these strategies primarily relied on individual volunteers instead of a formally paid and structured network [3].

In 2019, the country implemented a structured and paid (CHW) network in the three key foci of Greater Santo Domingo that reported 96% (1,243/1,292) of all national indigenous cases at the time: Los Tres Brazos, La Cienaga and San Cristobal. The goal of the malaria specific CHW network is to strengthen malaria detection and treatment through proactive household visits. CHWs are recruited based on community recognition, literacy, place of residency and other specific criteria. They were trained in a classroom setting, over five days with both theoretical and practical components. After training, CHWs were selected based on training completion and performance on posttest. For their first week of service, they were closely supervised in the field by the MOH personnel. CHWs are required to visit 100–200 houses per week, testing symptomatic people with rapid diagnostic tests and blood smears, and providing directly observed therapy for all three days of treatment for anyone testing positive.

The complete criteria and procedures for the design, implementation, and management of the CHW network are detailed in the National CHW Network Implementation and Management Guidelines [4].

Since CHW's implementation in 2019, urban malaria has nearly been eliminated and reported cases have increased in more rural provinces, such as San Juan (Fig 1). This research seeks to document the association between a structured CHW network and progress toward malaria elimination in the Dominican Republic. It quantifies the impact of COVID-19 disruptions on CHWs' performance and explores the ability of a well-supported CHW network to bounce back after routine operations have recommenced.

## Methods

The implementation and management of the Community Health Worker (CHW) network in the Greater Santo Domingo foci was led by the Ministry of Health's Centro de Prevención y Control de Enfermedades Transmitidas por Vectores y Zoonosis (CECOVEZ). CHW network data is securely stored on a SurveyCTO cloud server, version 2.8.1.1, which is compliant with GDPR. Data access is restricted to the Ministry of Health and select staff from the Clinton Health Access Initiative, who supported this project.

This study analyzed data from CHW implementation in La Cienaga (June 2019), San Cristobal (June 2019), and Los Tres Brazos (December 2019) through December 2022, including COVID-19 interruptions. CHW performance was assessed across three periods: (1) pre-

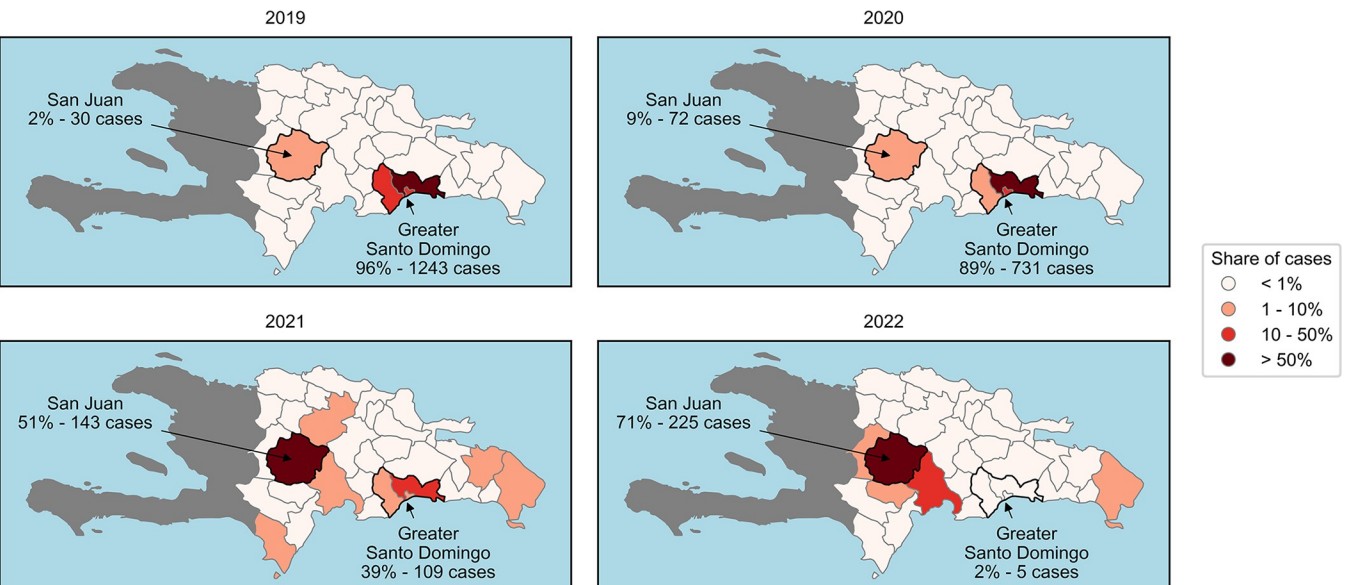

**Fig 1. Distribution of malaria cases in Dominican Republic's high transmission foci for the years 2019–2022.** The high transmission foci of Greater Santo Domingo and San Juan are highlighted. Data source: Centro de Prevención y Control de Enfermedades Transmitidas por Vectores y Zoonosis (CECOVEZ) national malaria risk stratification database. The basemap shapefile for the Dominican Republic was obtained from the Natural Earth project (link: https://www.naturalearthdata.com/about/terms-of-use/). The basemap shapefile for Haiti was sourced from the Humanitarian Data Exchange (HDX, link: https://data.humdata.org/dataset/cod-ab-hti/resource/9b57a285-e12f-4d1a-b167-676d96a2b4af). Terms of use: Haiti shapefiles are licensed under the Creative Commons Attribution 4.0 International license (CC BY 4.0); the Dominican Republic shapefiles are public domain map data provided by the Natural Earth project, available at naturalearthdata.com [5–7].

COVID-19 (until March 2020), (2) during COVID-19 (March 2020 to July 2021), and (3) post-COVID-19 (August 2021 to December 2022). During the COVID-19 lockdown (March-September 2020), government guidelines prohibited household visits entirely, halting all CHW activities. As restrictions eased (September 2020-February 2021), CHWs were permitted to make limited house visits, 2–3 days a week, depending on the availability of protective equipment. During this period, the number of household visits was affected by the willingness of community members to allow CHWs into their homes as well as the personal comfort level of CHWs in making household visits during the pandemic. After February 2021, restrictions were lifted, however, administrative delays in contract renewals further interrupted CHW's ability to work since they were not required to perform household visitations until July 2021 when contracts were renewed.

Two primary data sources were used. SurveyCTO, an online platform implemented to ensure CHWs monitoring and evaluation (M&E) [8], provided data on CHW performance, including number of houses visited per day, persons contacted, tests performed, and cases detected [9–12]. Weekly CHW performance evaluations began at the onset of implementing the network and have been ongoing since, resulting in 4,989 CHW M&E forms collected in the SurveyCTO database. The MoH collects malaria data using an internal database updated on a weekly basis by their epidemiology team. The MoH malaria database was used to review data on overall epidemiological trends and case management practices at other levels of the health system [13].

Results are presented across four categories: (iv) household visitation; (v) malaria testing; (vi) malaria case detection; and (vii) time between symptom onset and treatment. Whenever it was possible, data were assessed during each of the three time periods. Using Stata version 14.2, paired t-tests were used to compare average monthly household visits, rapid diagnostic

tests (RDTs) performed, and test positivity for those CHWs who were active in both periods of comparison (i.e., pre-during, during-posts, pre-post).

## Results

### CHW household visitation, testing and case detection

On average, during the pre-interruption phase, each CHW visited 130 houses per week. During the interruption phase, that number decreased to 80 houses per week before increasing to 185 houses per week in the post interruption phase (Fig 2).

Paired t-tests comparing mean weekly household visits for CHWs working pre-and-during interruption periods (t = 16.29; df = 58; p < 0.001), during and post interruption (t = -15.76, df = 46; p < 0.001), were found to be significantly different. During the pre-interruption time period, on average, CHWs complied with their target visitation (Table 1) by 98%. During the interruptions time period, CHWs performed 48% of the target household visitations. During the post interruption period, CHWs complied with 93% of their expected household visitation.

Table 1 shows the total number of household visitations per focus during the different time periods, highlighting the different visitation targets per focus and number of active CHWs.

On average, in all foci, during the pre-interruption time period, each CHW performed 7 RDTs per week. During interruptions, each contracted CHW performed an average of 2 RDTs per week. Post interruptions, CHWs performed an average of 3 RDTs per week. Paired t-tests comparing mean weekly RDTs performed by CHWs found significantly fewer RDTs performed during interruptions (2.0) compared to the same CHWs working pre interruption (7.8) (t = 12.52; df = 58; p < 0.001). For CHW working pre and post interruption (t = 9.35; df = 42; p = 0.0417) the number of RDTs dropped significantly from 8.3 to 3.1, respectively. The difference in mean monthly RDTs by CHWs increased significantly from 2.1 in the interruption compared to 3.1 post interruption (t = -4.73; df = 46; p < 0.001).

From 2019 to 2022 (over all three time periods combined), CHWs in all foci detected 36% (467/1,311) of cases, while other points of care (health facilities and local MoH personnel who perform active and passive case detection in their corresponding sectors) detected 64% (844/1,311). In the pre-interruptions time period CHWs detected 43% of cases (419/973), during interruptions this percentage decreased to 14% (45/321), and post interruptions increased to 18% (3/17) (Fig 2).

### Test positivity comparison

On average, across all foci, during the pre-interruption phase, CHW test positivity rate was 4%. During interruptions, test positivity rate decreased to 2% and post interruptions it continued to decrease to 0.04% The difference in mean monthly TPR for CHWs working pre and during interruption were not significantly different r (t = 1.843; df = 55; p = 0.0707) using a threshold of 0.05, unlike the mean monthly TPR during v post (t = 3.72; df = 43, p = 0.006) which was significant.

### Comparison of active vs. passive CHWs case detection

83% (389/467) of cases detected by CHWs were detected through active case detection (CHWs household visitations). The remaining 17% (78/467) percent of cases detected by CHWs were detected passively through symptomatic patients electing to seek out CHWs for malaria testing (Table 2).

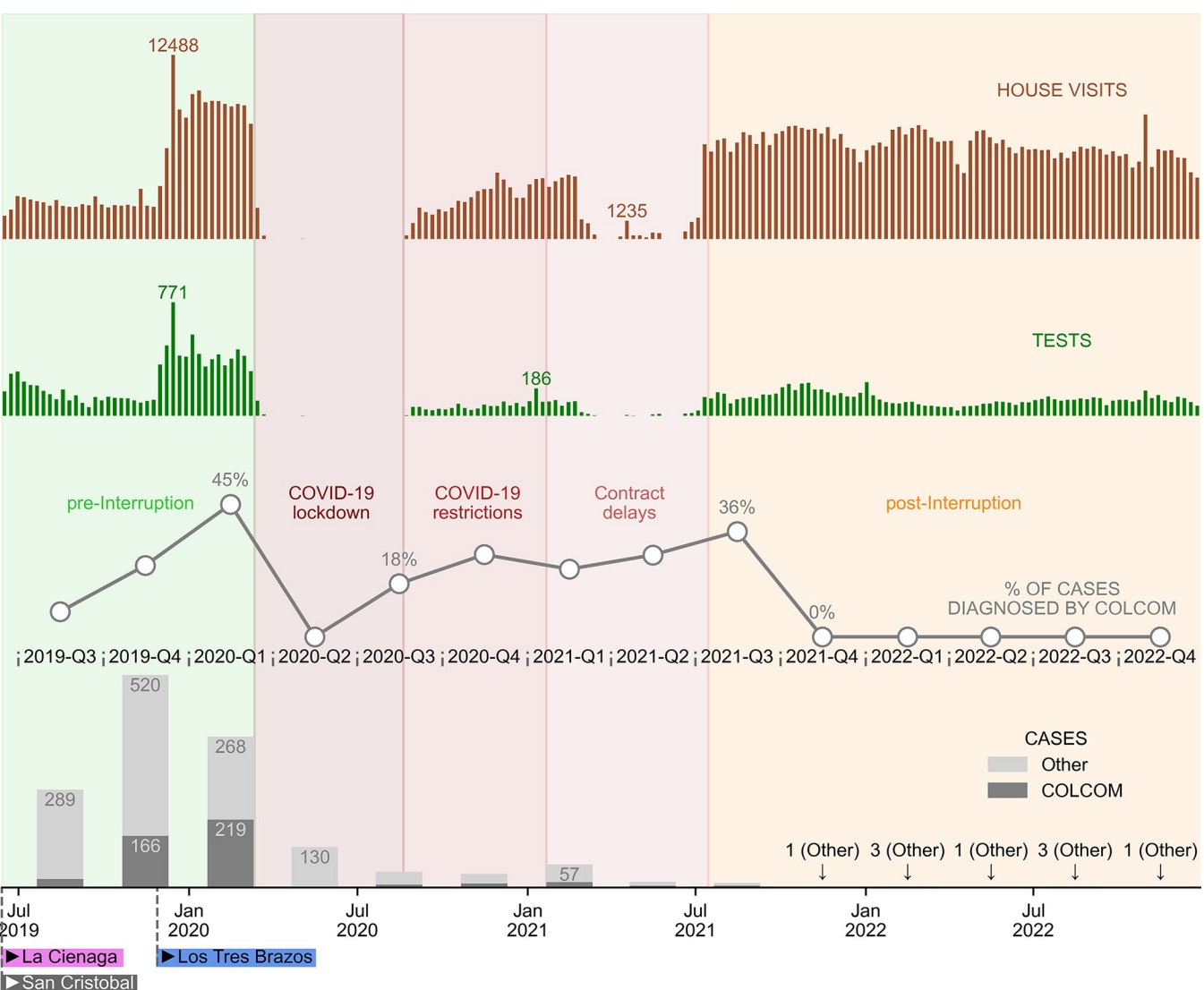

**Fig 2. Key CHWs' performance indicators by focus and time period.** Barchart representing the number of house visits and tests conducted weekly by CHWs in each of the 3 foci since the implementation of the CHW network. The bars (and line) plots represent, the total number (and corresponding percentage) of cases detected by CHWs compared to other cases detected by other entities (local MoH personnel who perform active case detection, and health facilities), by quarter, since the implementation of the CHW network until 2022. Background colors denote significant periods of time during which the performance of the network was affected. Bottom labels indicate the start of the CHW network implementation, for each focus. Data sources: SurveyCTO Weekly La Cienaga and San Cristobal Monitoring and Evaluation Database, SurveyCTO Weekly Los Tres Brazos Monitoring and Evaluation Database, Weekly La Cienaga and San Cristobal Monitoring and Evaluation Confirmed Case Database, Weekly Los Tres Brazos Monitoring and Evaluation Confirmed Case Database, Centro de Prevencion y Control de Enfermedades Transmitidas por Vectores y Zoonosis (CECOVEZ) Confirmed Cases Database [9–13].

Data sources: SurveyCTO Weekly La Cienaga and San Cristobal Monitoring and Evaluation Database, SurveyCTO Weekly Los Tres Brazos Monitoring and Evaluation Database [9, 10].

## Time between symptom onset and diagnoses

Of cases detected by CHWs, 28% were detected within WHO's recommended timeframe of 48 hours within symptom onset [2], compared to only 13.5 of cases diagnosed by other entities (health facilities and local level MoH). In the pre interruption period, 88/301 cases diagnosed

**Table 1. CHWs house visits.**

| Foci | Pre-Interruption LC/SC: June 2019—March 2020 LTB: December 2019—March 2020 | | | During Interruptions March 2020—July 2021 | | | Post Interruptions July 2021—December 2022 | | |
|---|---|---|---|---|---|---|---|---|---|
| | # CHWs | Average weekly house visits performed per CHW | Weekly house visits expected per CHW | # CHWs | Average weekly house visits performed per CHW | Weekly house visits expected per CHW | # CHWs | Average weekly house visits performed per CHW | Weekly house visits expected per CHW |
| Los Tres Brazos | 44 | 158 | 200 | 49 | 87 | 200 | 32 | 186 | 200 |
| La Cienaga | 20 | 111 | 100 | 14 | 54 | 100 | 11 | 187 | 200 |
| San Cristobal | 6 | 113 | 100 | 6 | 80 | 100 | 4 | 181 | 200 |

Data sources: SurveyCTO Weekly La Cienaga and San Cristobal Monitoring and Evaluation Database, SurveyCTO Weekly Los Tres Brazos Monitoring and Evaluation Database [9, 10].

by CHWs were diagnosed within the WHO recommended timeframe. During the interruption phase, 7 out of 34 cases were diagnosed within 2 days or less of symptom onset (21%), and post interruption only 1 out of 6 cases (17%) were detected within this time frame.

The median time between symptom onset and diagnosis for CHWs was 4 days during the pre-interruption period, compared to 5 days for other entities during the same period. During the interruptions period, the median time for both was 5 days. While very few cases were captured post interruption; the median time between symptom onset and diagnoses for those found by CHWs was 6.5 days compared to 7 days for other entities (Fig 3).

It is important to note that CHWs detected 467 cases during the study period, according to the SurveyCTO database used to manage CHWs' activities. However, only 341 of these cases could be matched in the CECOVEZ database. Therefore, 126 cases detected by CHWs were included with those identified by other entities, as we were unable to specifically identify those as being detected by CHWs. In the pre-interruption time period, 103 cases were not matched between both data bases, in the during interruption time period 21 cases were not matched, and in the post-interruption period 2 cases were not matched.

## Timeliness of CHWs treatment administration over time

WHO recommends that all patients are treated within 24 hours [2]. 96% of the cases detected by CHWs were treated within the first 24 hours. Pre-interruptions, Pre-interruptions, CHWs treated 392/409 cases within the first 24 hours, during interruption CHWs treated 53/55 cases in this time frame and post interruptions all (3/3) cases detected by CHWs were detected within this time frame. Time between diagnoses and treatment is not available for cases detected by other entities; hence, it is not possible to compare these data.

**Table 2. CHW confirmed cases by detection type and test positivity rate (TPR).**

| # CHWs | Rapid Diagnosed Tests (RDTs) performed by CHW | Total Cases detected by CHW | Active case detection by CHW (% total) | Passive case detection by CHW (% total) | Test Positivity Rate (TPR) |
|---|---|---|---|---|---|
| Pre-Interruption: June 2019—March 2020 | | | | | |
| 70 | 9932 | 419 | 346 (83%) | 73 (17%) | 4% |
| During Interruptions: March 2020—July 2021 | | | | | |
| 63 | 2487 | 45 | 40 (89%) | 5 (11%) | 2% |
| Post Interruptions: July 2021—December 2022 | | | | | |
| 47 | 8560 | 3 | 3 (100%) | 0 | 0.04% |

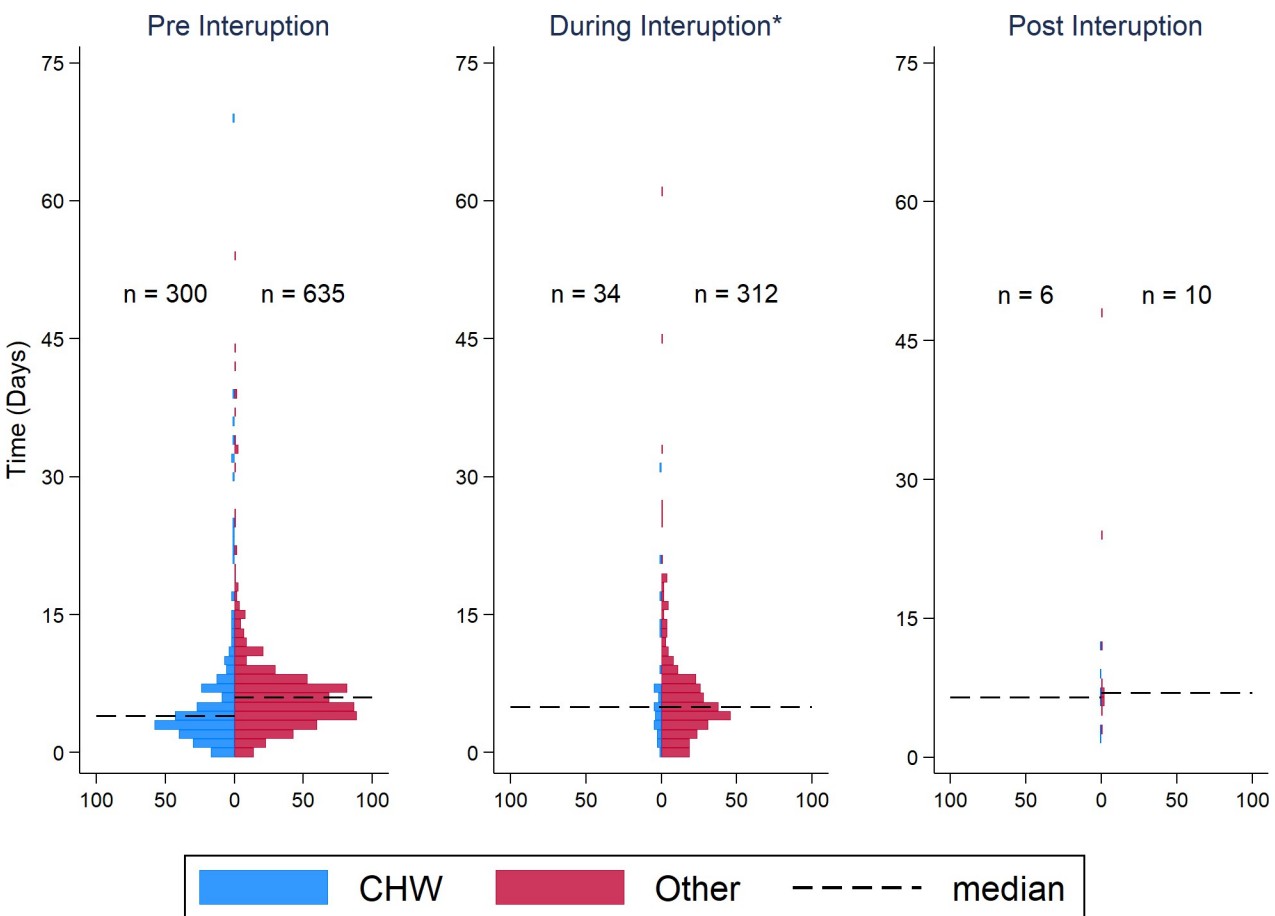

**Fig 3. Time between symptom onset and diagnosis.** Histogram charts illustrate the days between symptom onset and diagnosis (y axis) for cases diagnosed by CHW or other entity (counts on X axis). The dotted line represents the median value by provider type and period. *One outlier of 122 removed for Other for visualization purposes only. Data sources: SurveyCTO Weekly La Cienaga and San Cristobal Monitoring and Evaluation Database, SurveyCTO Weekly Los Tres Brazos Monitoring and Evaluation Database, Centro de Prevencion y Control de Enfermedades Transmitidas por Vectores y Zoonosis (CECOVEZ) Confirmed Case Database [9, 10, 13].

## Discussion

A structured CHW network of over 80 CHWs contributed to malaria case reductions in three urban foci of the DR by detecting a high proportion of all malaria cases (over 44% when working without interruptions) while reducing the time between symptom onset and diagnoses compared to other malaria care providers. Results align with other CHW implementation research findings and with WHO guidelines, which emphasize the importance of a structured community-based recruitment process and integration of CHWs within existing primary healthcare systems [14–17].

This investigation provides the first documentation in Central America and the Caribbean of the impact of COVID-19 on performance of an established malaria CHW network. COVID-19 interruptions are observed across a number of axes, including CHW's household visitation, testing rate, case detection and timeliness of case management services. The impact of COVID-19 interruptions was observed as decreases in CHWs: (I) average number of weekly household visits (46% decrease), (II) average number of weekly RDTs performed by each CHW (from 7 to 2), (III) number of cases detected by the CHW network (89% decrease) and (VI) compliance with WHO's recommended timeframe for time between symptom onset and

case detection (from 29% compliance to 21% compliance). Despite these secondary effects observed during COVID-19 interruptions, our study indicates that a well-structured and well-compensated CHW program can bounce back quickly from such shocks as demonstrated by rapid rebounds in retention, reporting levels and the number of households visited in the post-interruption time period.

Proactive malaria case detection has shown promising outcomes, including heightened case detection rates, and expanded treatment opportunities in numerous countries [3, 18–21], but its application has been minimal in large-scale CHW programs in Latin America. Our work demonstrates that there is high potential for proactive household visitation by CHWs, when such daily work is integrated into their responsibilities from program inception and pay is reflected of the work they are contracted to do. Over the course of this study, 83% of CHW-detected malaria cases were detected through proactive household visitation. The ability of CHWs to reduce the time between symptom onset and diagnosis, especially pre-interruptions (both passively and proactively), may help cut transmission and catalyze malaria elimination. However, over 75% of the total number cases detected by both CHWs and other service providers were detected after 24 hours of symptom onset. This delay may be attributed to various factors, including inadequate access to healthcare services and CHWs. Moreover, it underscores a potential need for behavioral change within the population, as individuals might not be actively seeking timely diagnosis.

In addition to yielding malaria-specific benefits, high rates of routine encounters between CHWs and households provide a strong foundation for expanding into other areas of health work, which may in turn increase demand for malaria testing and treatment services as burden continues to decline [18]. Over the course of the three time periods, malaria RDTs were performed in fewer than 5% of CHW household visits. If 95% of household visits conducted by CHWs do not include an RDT, much could be gained from a health system and community trust perspective by equipping the CHWs with complementary skills and services such as TB, maternal and child health, vaccinations, and health promotion. Previous studies have emphasized how integrating different tasks to vertical CHW program can increase care seeking and confirmatory malaria testing rates [3, 22]. If CHWs are fairly paid, and their catchment populations carefully and reasonably calculated and locally validated, integrating household visitation into their routine work is sensible and should come without an extra cost [23].

Our evaluation has limitations. Lack of data from other service providers prevented more comparisons between service provider type (health posts vs CHWs). More generally, CHWs were selected to work the in the most endemic areas of the urban foci, thereby rendering it impossible to have comparable sites with no CHWs to make stronger inference on their effect in reducing cases and/or the effect of COVID-19 lockdowns on transmission. It is unclear to what effect stigma or fear related to reporting a fever during the COVID-19 pandemic might have affected our findings. Our results are based on a relatively small, paid, malaria-only CHW cadre which may decrease our study's generalizability to other large-scale volunteer CHW networks in more endemic settings or that rely on volunteer workers. This study does not include an economic evaluation which precludes us from quantifying the cost per case averted from a paid CHW network.

## Conclusion

This study adds to the existing research that a well-organized CHW network, with defined roles and management protocols, holds the potential to undertake a significant portion of malaria case management services, increase early malaria case detection, and contribute substantially to elimination of local malaria transmission. These findings complement an

expanding body of research highlighting the viability and advantages of CHW-led proactive household visits. Despite disruptions such as the COVID-19 pandemic, a well-structured CHW program can uphold and further enhance the gains achieved.

## Acknowledgments

We would like to thank the Community Health Workers for their hard work and consistency. The local and central level health authorities for their dedication and commitment with the community. We particularly acknowledge the following personnel and their teams for their efforts implementing and managing the CHW network: Mr.Domingo Cabral, Dr. Francisco Camilo, Dr. Carmen Rosa Then, Dr. Josvane Japa, Dr. Alba Núñez, Dr. Alondra Frías, Dr. Juan David Domínguez Gálvez, Dr. Diana Taveras, Dra. Mirna Salomón, Dr. Yeinmy Campusano.

## Author Contributions

**Conceptualization:** Nicole Michelén Ströfer, Jose Luis Cruz Raposo.

**Data curation:** Nicole Michelén Ströfer, Natalia Tejada Bueno.

**Formal analysis:** Justin T. Lana.

**Investigation:** Natalia Tejada Bueno, María Yinet Santos Félix.

**Methodology:** Nicole Michelén Ströfer.

**Visualization:** Nicole Michelén Ströfer, Natalia Tejada Bueno, Justin T. Lana, Valérian Turbé.

**Writing – original draft:** Nicole Michelén Ströfer, Natalia Tejada Bueno, Justin T. Lana, Valérian Turbé, Harriet G. Napier.

**Writing – review & editing:** Nicole Michelén Ströfer, Natalia Tejada Bueno, María Yinet Santos Félix, Jose Luis Cruz Raposo, Justin T. Lana, Valérian Turbé, Harriet G. Napier, Justin M. Cohen.

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
