## [Decision Letter · Decision Letter 0]

18 Oct 2024

PGPH-D-24-01808

The role of a structured community health worker network in achieving malaria elimination goals in the Dominican Republic: an impact evaluation amid COVID-19 disruptions

Dear Dr. Michelen Strofer,

Thank you for submitting your manuscript to PLOS Global Public Health. After careful consideration, we feel that it has merit but does not fully meet PLOS Global Public Health’s publication criteria as it currently stands. Therefore, we invite you to submit a revised version of the manuscript that addresses the points raised during the review process.

We look forward to receiving your revised manuscript.

Kind regards,

Nirbhay Kumar, PhD

Professor, Global Health

Academic Editor

Journal Requirements:

Additional Editor Comments (if provided):

Reviewers' comments:

Reviewer's Responses to Questions

**Comments to the Author**

1. Does this manuscript meet PLOS Global Public Health’s publication criteria? Is the manuscript technically sound, and do the data support the conclusions? The manuscript must describe methodologically and ethically rigorous research with conclusions that are appropriately drawn based on the data presented.

Reviewer #1: Yes

2. Has the statistical analysis been performed appropriately and rigorously?

Reviewer #1: Yes

3. Have the authors made all data underlying the findings in their manuscript fully available (please refer to the Data Availability Statement at the start of the manuscript PDF file)?

Reviewer #1: Yes

4. Is the manuscript presented in an intelligible fashion and written in standard English?

Reviewer #1: Yes

5. Review Comments to the Author

Reviewer #1: Well written paper

Comments:

#1 - Characteristics of Covid lockdown - It would be helpful to have more detail on government guidelines during the Covid lockdown. If a household visit does not take place, possible reasons are 1) government guidelines prohibiting household visits, 2) government allows household visits, but CHWs prefer to not make them, 3) CHWs make household visits, but householders do not allow CHWs to visit / to enter the house. If there were government prohibitions on household visits, on what date did they come into effect, and when did they end?

#2 - Characteristics of CHW program - The paper tells us almost nothing about the CHWs, or the larger CHW program. What other responsibilities do CHWs have? Are they malaria-specific, or do they have a range of responsibilities and assigned tasks? Do they report to the malaria program, or to a national CHW program? How are they recruited? What training do they receive (e.g., 2 weeks of basic training in classroom followed by 4 weeks of close supervision)?

6. PLOS authors have the option to publish the peer review history of their article (what does this mean?). If published, this will include your full peer review and any attached files.

**Do you want your identity to be public for this peer review?** For information about this choice, including consent withdrawal, please see our Privacy Policy.

Reviewer #1: No

---

## [Editor Report · Decision Letter 1]

6 Nov 2024

The role of a structured community health worker network in achieving malaria elimination goals in the Dominican Republic: an impact evaluation amid COVID-19 disruptions

PGPH-D-24-01808R1

Dear Ms. Michelen Strofer,

We are pleased to inform you that your manuscript 'The role of a structured community health worker network in achieving malaria elimination goals in the Dominican Republic: an impact evaluation amid COVID-19 disruptions' has been provisionally accepted for publication in PLOS Global Public Health.

Best regards,

Nirbhay Kumar

Academic Editor